# Influential Factors in Transportation and Mechanical Properties of Aeolian Sand-Based Cemented Filling Material

**Nan Zhou [1] , Haobin Ma [1], Shenyang Ouyang [1],*, Deon Germain [1] and Tao Hou [2]**

[1] State Key Laboratory of Coal Resources and Safe Mining, School of Mines,
  China University of Mining and Technology, Xuzhou 221116, China;
  zhounanyou@126.com (N.Z.); cumtmhb@163.com (H.M.); germaindeon@live.fr (D.G.)
[2] Shandong Tang Kou Coal Industry Co., Ltd., Jining 272055, China; ouyang@cumt.edu.cn
* Correspondence: ouyangshenyang@126.com; Tel.: +86-17851147585

**Abstract:** Given that normal filling technology generally cannot be used for mining in the western part of China, as it has only a few sources for filling gangue, the feasibility of instead using cemented filling materials with aeolian sand as the aggregate is discussed in this study. We used laboratory tests to study how the fly ash (FA) content, cement content, lime–slag (LS) content, and concentration influence the transportation and mechanical properties of aeolian-sand-based cemented filling material. The internal microstructures and distributions of the elements in filled objects for curing times of 3 and 7 days are analyzed using scanning electron microscopy (SEM) and energy-dispersive spectroscopy (EDS). The experimental results show that: (i) the bleeding rate and slump of the filling-material slurry decrease gradually as the fly ash content, cement content, lime–slag content, and concentration increase, (ii) while the mechanical properties of the filled object increase. The optimal proportions for the aeolian sand-based cemented filling material include a concentration of 76%, a fly ash content of 47.5%, a cement content of 12.5%, a lime–slag content of 5%, and an aeolian sand content of 35%. The SEM observations show that the needle/rod-like ettringite (AFt) and amorphous and flocculent tobermorite (C-S-H) gel are the main early hydration products of a filled object with the above specific proportions. After increasing the curing time from 3 to 7 days, the AFt content decreases gradually, while the C-S-H content and the compactness increase.

**Keywords:** green mining; cemented filling; transportation property; mechanical property

## 1. Introduction

Ground subsidence, water and soil loss, and the protection of the ecological environment are some of the primary challenges facing the coal mining industry [1–3]. In China, with the western part of the country gradually becoming the main coal mining region, emphasis has been placed on ensuring that the mining occurs in a sensible manner in fragile arid and semi-arid regions. As an important means of realizing green coal mining, the associated filling technology can effectively balance the economic benefits with the protection of the ecological environment [4,5]. However, the western part of China has only a few sources for filling gangue, due to the special deposits and geological conditions of its mining areas, thereby making normal filling technologies infeasible.

A considerable amount of research has recently been conducted on filling materials (coal gangue, fly ash, and construction waste) and microstructural characteristics for coal mines [6–8]. Feng et al. [9] studied the mechanical properties of cemented filling materials in which gangue was partially replaced by waste concrete. They found that when the replacement ratio of waste concrete was 37%, the filled

object had optimal mechanical properties. Ercikdi et al. [10] partially replaced ordinary Portland cement with industrial waste (e.g., slag and FA) and investigated how the type of waste affected the early strength and later strength of the filled object. Liu et al. [11] established the damage constitutive equation of a filled object for which tailings were used as aggregates by analyzing the deformation and failure characteristics. Wu et al. [12] analyzed the effect of particle size distribution on the strength characteristics of cemented filling materials. Huynh et al. [13] explored the effect of polyphosphate and naphthalene sulfonate formaldehyde condensates on the rheological properties of dewatered tailings and cemented filling materials. Deng et al. [14] developed a new type of cemented filling material, using waste rock as a coarse aggregate, FA as a fine powder, slag as an activator, and ordinary Portland cement as a binder. Cao et al. [15] analyzed the effects of solid content, cement-to-tailings ratio, and curing time on the strength of tailing-cemented filling materials. Xu et al. [16] studied the influence of different types of binders and their amounts on the microstructure of the filled object, which allowed them to obtain the best binder type and microstructure change law. Taheri and Tatsuoka [17,18] analyzed the small- and large-strain behavior of a cement-treated soil during various loading histories and testing conditions. Niroshan et al. [19] determined the relationship between microstructure and the long-term mechanical properties of cemented filling materials.

Based on the above, the previous research results mainly aimed to use construction waste and filling materials using tailings as aggregates. However, cemented filling material with aeolian sand—which largely covers the ground surface in the mining area of Western China—as the aggregate has not been studied yet. For the present study, we conducted an analysis of the microstructure and macroscopic mechanical properties of AS-based cemented filling materials using laboratory proportioning. The feasibility of using such filling materials is discussed and the obtained optimal proportions of filling materials that are suitable for the mining region of Western China are demonstrated.

## 2. Materials and Methods

### 2.1. Materials

The main experimental materials used in the present study are AS, FA, cementing material, and additives. The AS was obtained from local sources in the Yulin region of the Shaanxi Province, China. The FA was made up of combustion waste from a power plant. The cementing material was grade 42.5 ordinary Portland cement (OPC). A small amount of lime–slag was added during the experiments as an activator of the FA.

#### 2.1.1. Aeolian Sand

AS was obtained from the local mining area in Yulin in Northwest China, where it is cheap to buy and distributed naturally, widely, and in massive amounts. Figure 1 shows the microstructures observed through scanning electron microscopy (SEM). It was found that there were abundant small voids among the particles. The particles were not angular with irregular or rough surfaces and they did not contain cracks or pores. Figure 2 shows the distribution of grain sizes. The AS particles were fine and hard with a main grain size of 100–300 μm and a minimum particle size of 50 μm, thereby allowing good pressure transfer. The coefficient of uniformity and coefficient of curvature of the AS were 1.8–2.0 and 0.9–1.1, respectively.

The chemical composition of the AS sample was tested with a D8 ADVANCE X-ray diffraction (XRD) instrument (Bruker, Karlsruhe, Germany), as shown in Figure 3 and Table 1. The AS was shown to contain a considerable amount of quartz and feldspars, as well as some calcites and illite. $SiO_2$ and $Al_2O_3$ were the main oxides within the AS (67.49% and 8.75% by weight, respectively).

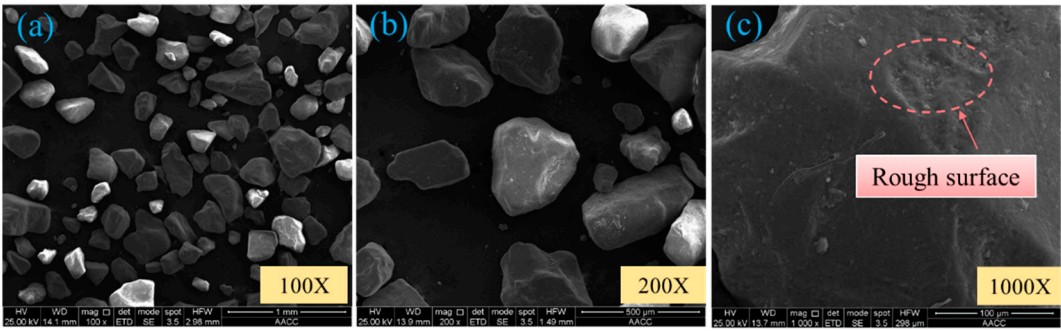

**Figure 1.** Scanning electron microscopy (SEM) micrographs of AS. Magnification levels are shown in the bottom right-hand corners: (**a**) AS at 100x magnification; (**b**) AS at 200x magnification; (**c**) AS at 1000x magnification.

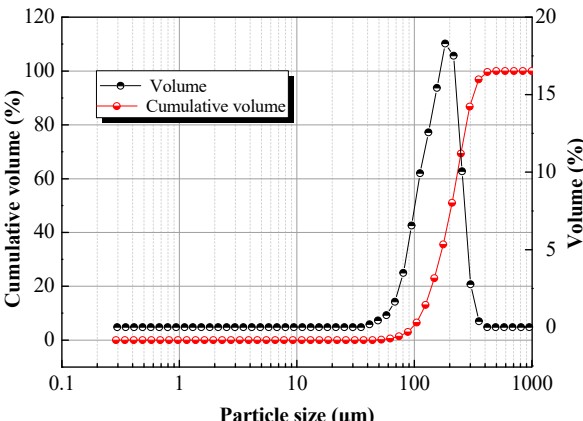

**Figure 2.** Particle size distribution of AS.

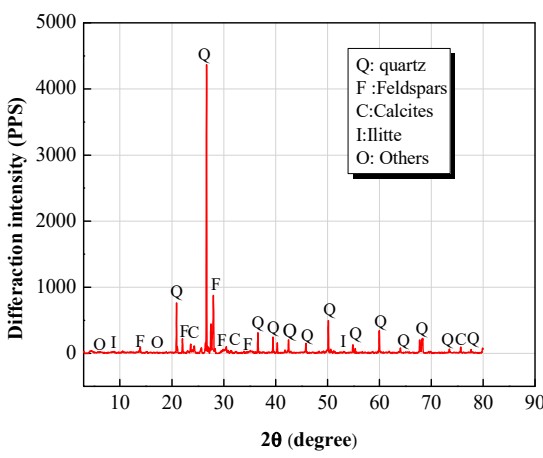

**Figure 3.** X-ray diffraction (XRD) pattern of AS.

**Table 1.** Chemical components of the studied materials, in wt. %. AS—Aeolian sand; FA—Fly ash.

| Samples | $Na_2O$ | MgO | $Al_2O_3$ | $SiO_2$ | $K_2O$ | CaO | $Fe_2O_3$ | Others |
|---------|---------|------|-----------|---------|--------|------|-----------|--------|
| AS | 2.22 | 1.3 | 8.75 | 67.49 | 2.86 | 5.78 | 4.35 | 7.25 |
| FA | 1.31 | 1.62 | 26.69 | 41.29 | 0.94 | 8.73 | 5.21 | 14.21 |

### 2.1.2. Fly Ash

FA was obtained from a coal-fired power plant near the mining area in Yulin. FA is gray/black in color and is similar to cement in appearance. Figure 4 shows the microstructures obtained through SEM, and Figure 5 shows the distribution of particles. The range of the main grain size of the FA was

10–100 μm, and the minimum particle size was 3.5 μm, with 65 ± 5% of the FA having grain sizes less than 50 μm. When the FA was piled up, it was in a powdered form with a fluffy interior and had high plasticity. The FA particles were found to be irregular chips and cohesive objects under SEM observation. Under a greater SEM magnification, many glass beads with smooth and compact surfaces and grain sizes 1–10 μm were observed to be independent from or gathered together with other glass beads. The chemical composition of the FA sample is shown in Table 1; the main oxides of FA are $SiO_2$ and $Al_2O_3$, with percentages by weight of 41.29% and 26.69%, respectively. Those compositions play an important role in determining strength [20].

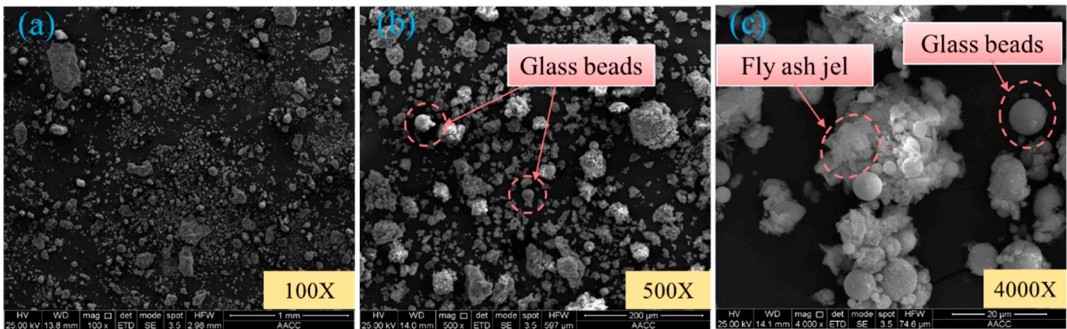

**Figure 4.** SEM micrographs of FA. Magnification levels are shown in the bottom right-hand corners: (**a**) FA at 100x magnification; (**b**) FA at 500x magnification; (**c**) FA at 4000x magnification.

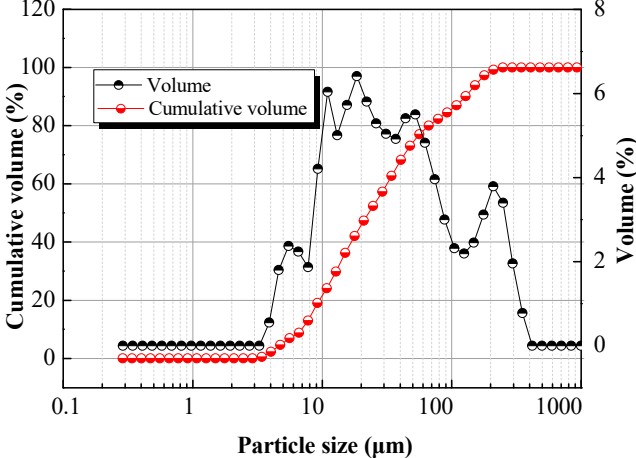

**Figure 5.** Particle size distribution of FA.

### 2.1.3. Cement

The cementing material used in the present experiments was grade 42.5 ordinary Portland cement, which had fine particles with main grain sizes in the range 30–60 μm and a minimum particle size of 1.7 μm. Figure 6 shows the distribution of particles. The grain sizes of 80 ± 5% of the cement particles were less than 60 μm, and approximately 50% of the cement particles had grain sizes of less than 30 μm.

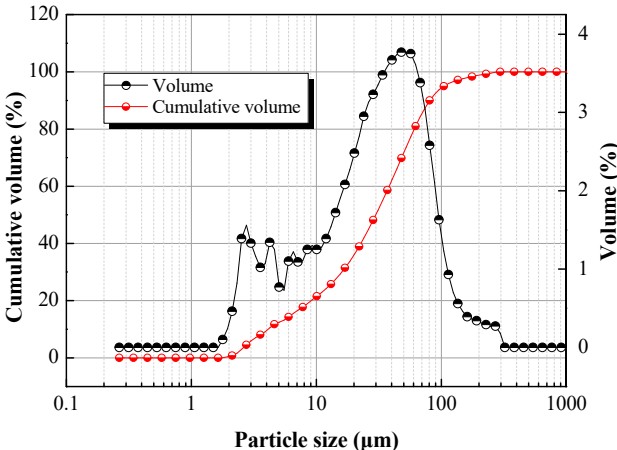

**Figure 6.** Distributions of particles of cement.

### 2.1.4. Lime–Slag

A small amount of LS was added during the experiments as an activator of the FA. The main ingredient of LS is calcium oxide, which is an air-hardening cementing material that has a poor cementing capacity in a water environment. The main purposes of adding LS are to excite FA activity [21,22]. Meanwhile, the calcium oxide in the LS can react with the water, thereby consuming free water and reducing the bleeding rate of the slurry.

### 2.2. Cemented Filling Material Preparation and Performance Test

To study the effects of the FA content, cement content, LS content, and concentration, four groups were designed by mixing AS, FA, cement, LS, and water in different proportions, as shown in Table 2. Group A represents the mixtures with different FA proportions, Group B represents the mixtures with different cement proportions, Group C represents the mixtures with different LS proportions, and Group D represents the mixtures with different concentration (concentration in this paper means solid content).

**Table 2.** Scheme of proportioning tests.

| Test Group | Test No. | Concentration (Solid Content) [wt. %] | Water Content [wt. %] | Solids Content Composition [wt. %] | | | |
| --- | --- | --- | --- | --- | --- | --- | --- |
| | | | | **Fly Ash** | **Cement** | **Lime Slag** | **Aeolian Sand** |
| **A** | **A1** | **72** | 28 | 42.5 | 12.5 | 5 | 40 |
| | A2 | 72 | 28 | 47.5 | 12.5 | 5 | 35 |
| | A3 | 72 | 28 | 52.5 | 12.5 | 5 | 30 |
| B | B1 | 72 | 28 | 47.5 | 7.5 | 5 | 40 |
| | B2 | 72 | 28 | 47.5 | 10 | 5 | 37.5 |
| | B3 | 72 | 28 | 47.5 | 12.5 | 5 | 35 |
| C | C1 | 72 | 28 | 47.5 | 12.5 | 0 | 40 |
| | C2 | 72 | 28 | 47.5 | 12.5 | 5 | 35 |
| | C3 | 72 | 28 | 47.5 | 12.5 | 10 | 30 |
| D | D1 | 72 | 28 | 47.5 | 12.5 | 5 | 35 |
| | D2 | 74 | 26 | 47.5 | 12.5 | 5 | 35 |
| | D3 | 76 | 24 | 47.5 | 12.5 | 5 | 35 |

After mixing the AS, FA, cement, and LS in certain proportions (wt. %), water was added to the mixture, which was subsequently stirred with a mechanical mixer into a fluidized slurry of cemented filling material. Part of the slurry was used to test the bleeding rate and slump according to the GB/T50080-2016 Chinese national standard for test method of performance on ordinary fresh concrete (China Academy of Building Research). The remaining slurry was used to make a specimen in a cubic steel mold with dimensions of 70.7 mm × 70.7 mm × 70.7 mm. This mold was removed after 8 h.

The specimens were then cured under standard curing conditions at room temperature of $20 \pm 2$ °C and a humidity of 95% for curing times of 1, 3, 7, 14, and 28 days. The mechanical properties of a filled object were tested for each curing time according to the GB/T23561.12-2010 methods for determining the physical and mechanical properties. Table 2 lists the scheme of the proportioning tests and Figure 7 shows the process of cemented filling material preparation and test.

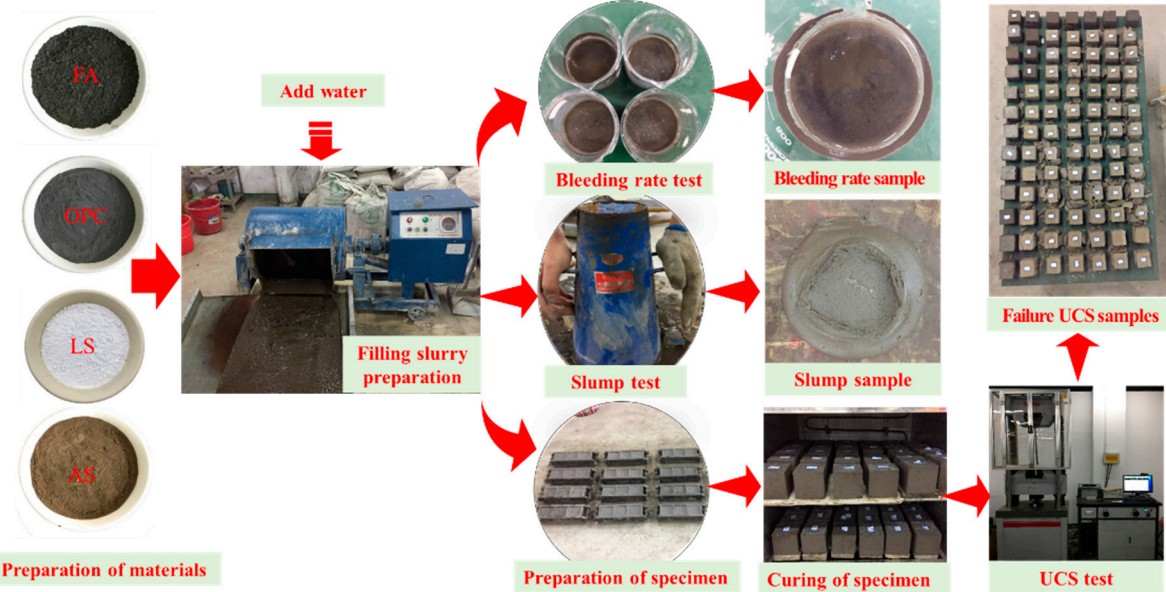

**Figure 7.** The process of cemented filling material preparation and testing. OPC—Ordinary Portland cement; LS—Lime–slag; UCS—Uniaxial compressive strength.

### 2.3. Microstructural Analysis

The microstructure of the filling material has an important influence on its strength. In this paper, a Quanta 250 scanning electron microscope (FEI, Hillsboro, Oregon state, USA) and a QUANTAX 400-10 energy-dispersive spectrometer (Bruker, Karlsruhe, Germany) were used to observe the microstructure and element distribution characteristics in the filled object. The observation sample is required to cut a rectangular parallelepiped with a bottom area of 10 mm × 10 mm and a height of 3–5 mm from the inside of the filling object. This was dried with a drying oven at a temperature of 60 °C for 24 h. The SEM and EDS scanning sample preparation process and analysis equipment are shown in Figure 8.

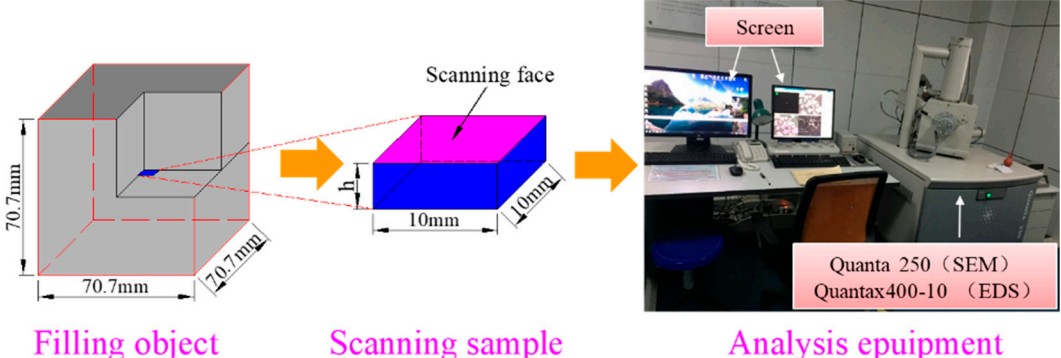

**Figure 8.** The SEM and EDS scanning sample preparation process and analysis equipment.

## 3. Results and Discussion

### 3.1. Analysis of Factors Affecting Macroscopic Properties of Cemented Filling Materials

3.1.1. Effects of Fly Ash Content on Properties of Filling Material

The proportioning tests were performed on Group A to investigate how the FA content affects the transportation and mechanical properties of the filling material. The mixtures in Group A had a concentration of 72%, a Portland cement content of 12.5%, and an LS content of 5%; the remaining 82.5% of the solid material was composed of FA and AS. There were three different content levels of FA, specifically 42.5%, 47.5%, and 52.5%.

Effect of Fly Ash Content on Transportation Properties of Filling Material

Figure 9a shows that increasing the FA content caused the bleeding rate and slump of the filling material slurry to decrease gradually. One reason for this was that when the FA came into contact with water, its surface adsorbed a layer of the water film. A higher FA content resulted in more free water being consumed to form the water film. Another reason for the effect of the FA content on the bleeding rate and slump was that, due to their fine gradation and homogenized deflocculating effect, the tiny glass beads in the FA reacted with water and generated flocks that became suspended in the filling material slurry. This hindered the subsidence of large aggregates, thereby enhancing the workability of the slurry. These decreases in the amount of free water and the subsidence speed of large aggregates gradually reduced the bleeding rate and slump of the slurry, the influence of which became more obvious as the FA content increased.

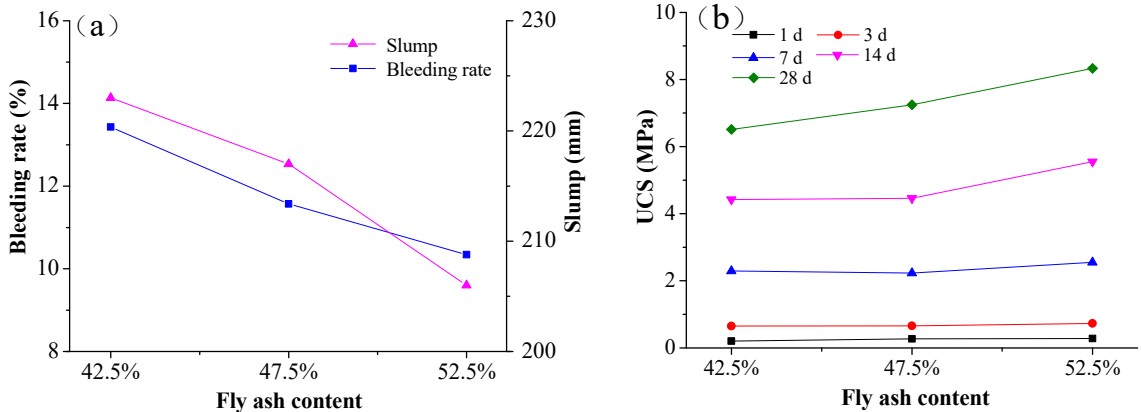

**Figure 9.** Effects of FA content on properties of filling material: (**a**) effects of FA content on transportation properties of filling material; (**b**) effects of FA content on mechanical properties of filling material.

Effect of Fly Ash Content on Mechanical Properties of Filling Material

Figure 9b shows that increasing the FA content caused the uniaxial compressive strength of the filling material to increase gradually, the effect of which was greater on the later strength of the filled object than on the early strength. After mixing and stirring the filling material, the tiny FA particles filled the gap of the filled object and thus enhanced its compactness. Meanwhile, there was an abundance of glass beads in the FA, and these participated in the hydration reaction of the cement by combining with $Ca(OH)_2$ during the hydration process. This gradually generated the hydration product with a certain cementing capacity (such as C-S-H), which reinforced the cementing among the aggregates and increased the strength of the filled object [23,24]. Since the hydration period of the FA was as long as 28 days, this increase was mainly in the later strength of the filled object [25].

### 3.1.2. Effects of Cement Content on Properties of Filling Material

The proportioning tests were performed on Group B to investigate how the cement content affected the transportation and mechanical properties of the filling material. The mixtures in Group B had a concentration of 72%, an FA content of 47.5%, and an LS content of 5%, and the remaining 47.5% of the solid materials were composed of cement and AS. There were three different content levels of cement, specifically 7.5%, 10%, and 12.5%.

Effect of Cement Content on Transportation Properties of Filling Material

Figure 10a shows that increasing the cement content caused the bleeding rate and slump of the filling material slurry to decrease gradually. When the cement content was 12.5%, the bleeding rate reached its minimum of 11.57% and the slump remained good (210 mm). According to the results from a laser particle analyzer, the main grain size of the cement used in the testing ranged from 30 to 60 μm. The cement particles wrapped up free water after making contact with water and then formed bound water. A higher cement content resulted in more free water being consumed [26], thereby increasing the viscosity of the slurry while decreasing its bleeding rate and slump.

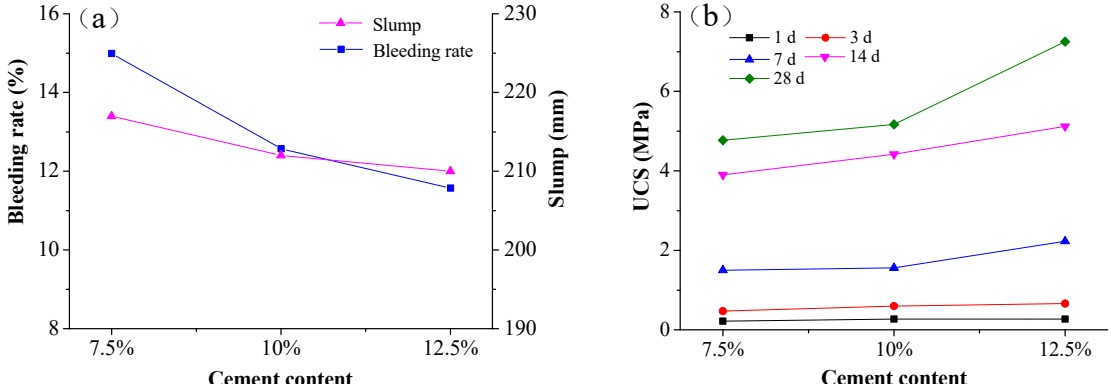

**Figure 10.** Effects of cement content on properties of filling material: (**a**) effects of cement content on transportation properties of filling material; (**b**) effects of cement content on mechanical properties of filling material.

Effect of Cement Content on Mechanical Properties of Filling Material

Figure 10b shows that increasing the cement content caused the uniaxial compressive strength of the filled object to increase, the effect of which was smaller on the 1- and 3-day strengths while being larger on the later strength for a curing time of 7 days or more. Among the filled objects with the same cement content, their uniaxial compressive strengths increased gradually with increased curing time. For instance, considering the filled objects with a cement content of 12.5%, the uniaxial compressive strengths of specimens cured for 1, 3, 7, 14, and 28 days were 0.27, 0.66, 2.23, 5.12, and 7.25 MPa, respectively. As the primary cementing material, cement played a leading role in cementing the filled object. Adjacent sand particles were cemented together by ettringite (AFt), C-S-H-generated or other hydration products during the cement hydration reaction, forming a sand–hydration-product composite structure with a certain load-bearing capacity [27,28]. The cementing effect increased gradually as the cement hydration proceeded, and there was more cementing material as the cement content increased. The results indicate that the strength of the filled object increased gradually as the cement content and curing time increased. When the cement content was 12.5%, the slump remained good, the bleeding rate reached its minimum, and the strengths for the various curing times reached their maxima. This showed that 12.5% was the optimal cement content.

### 3.1.3. Effects of Lime–Slag Content on Properties of Filling Material

The proportioning tests were performed on Group C to investigate how the LS content affected the transportation and mechanical properties of the filling material. The mixtures in Group C had a concentration of 72%, an FA content of 47.5%, and a Portland cement content of 12.5%; the remaining 40% of the solid materials were composed of LS and AS. There were three different content levels of LS, specifically 0%, 5%, and 10%.

Effect of Lime–Slag Content on Transportation Properties of Filling Material

Figure 11a shows that, as the LS was added, the bleeding rate and the slump of the filling material slurry both decreased, the effects of which became larger as the LS content increased. When 5% and 10% of LS were added, the bleeding rate decreased by 2.03% and 3.96%, respectively, while the slump decreased by 8 mm and 13 mm, respectively. This is because the water absorption capacity of the CaO contained in the LS reduced the free water content in the filling material slurry, causing the bleeding rate and the slump of the filling material slurry to decrease gradually.

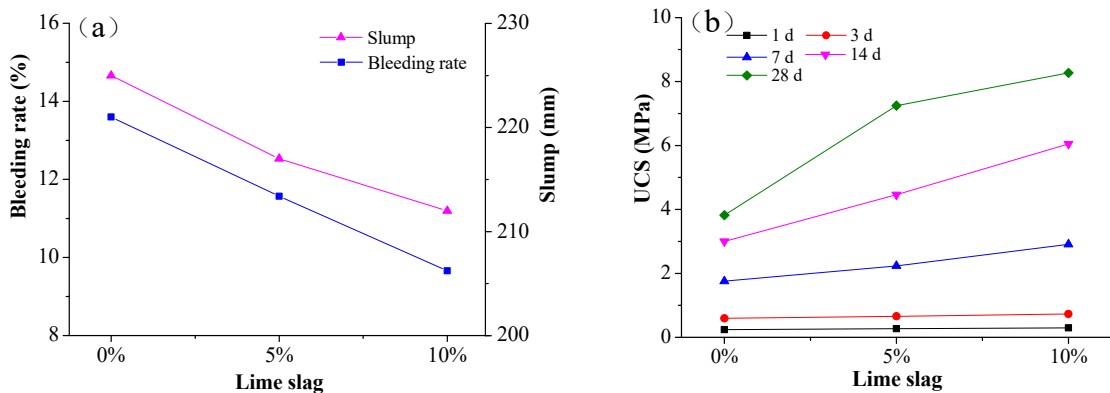

**Figure 11.** Effects of LS content on properties of filling material: (**a**) effects of LS content on transportation properties of filling material; (**b**) effects of LS content on mechanical properties of filling material.

Effect of Lime–Slag Content on Mechanical Properties of Filling Material

Figure 11b shows that, relative to an LS content of 0%, the LS contents of 5% and 10% increased the early strength (1 day) by 0.03 MPa and 0.06 MPa, respectively, and increased the later strength (28 days) by 3.43 MPa and 4.45 MPa, respectively. This indicates that adding the LS increased the strength of the filled object. This effect was enhanced by the addition of LS content and had a greater impact on the later strength. The excitation function of the hydration between the $Ca(OH)_2$ generated during hydration of the LS and the glass beads in the FA enhanced the hydration progress of the FA [29–31]. The effect of the LS on the filled object was mainly reflected in the later strength due to the long hydration time of the FA. The variation amplitudes in the 28-day uniaxial compressive strength show that the effect on the 28-day strength of the filled object of adding an LS content of 5% was similar to that of adding an LS content of 10%. Therefore, the added LS content can be determined as 5%.

### 3.1.4. Effects of Concentration on Properties of Filling Material

The proportioning tests were performed on Group D to investigate how the concentration affected the transportation and mechanical properties of filling material with the same solid components. The mixtures in Group D had an FA content of 47.5%, a Portland cement content of 12.5%, an LS content of 5%, and an AS content of 35%. There were three different levels of concentration, specifically 72%, 74%, and 76%.

Effect of Concentration on Transportation Properties of Filling Material

Figure 12a shows that with an increase in the concentration, the bleeding rate and slump of the filling-material slurry decreased significantly. When the concentration was 76%, the bleeding rate reached a minimum of 4.83%, which was smaller than the bleeding rate by 58.3% and 40.4% when the concentration was 72% and 74%, respectively. The slump decreased from 217 mm to 177 mm when the concentration increased from 72% to 76%. The subsidence process of large particles inside the filling material slurry was resisted by surface shear stress and suspended micro particles. When there was less water, the sinking resistance of the particles was larger; thus, the subsidence was slower and the bleeding rate was smaller [32,33]. Materials with the same proportion of solid particles had the same ability to absorb water. A material slurry with a larger concentration contained less free water, indicating that when the concentration was larger, the bleeding rate and slump were smaller.

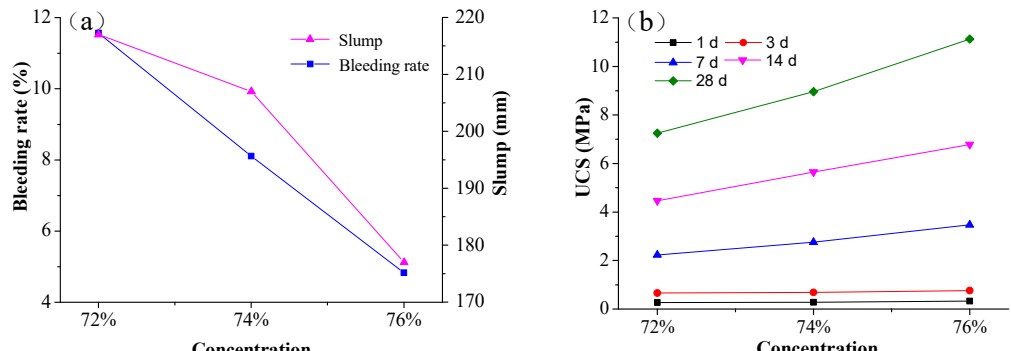

**Figure 12.** Effects of concentration on properties of filling material: (**a**) effects of concentration on transportation properties of filling material; (**b**) effects of concentration on mechanical properties of filling material.

Effect of Concentration on Mechanical Properties of Filling Material

Figure 12b shows that increasing the concentration enhanced the strength of filled objects with the same curing time, the effect of which became more obvious as the curing time increased. For instance, considering specimens for curing times of 1 and 28 days, relative to specimens with a concentration of 72%, the 1-day strength of specimens with a concentration of 74% increased by only 3.7%, whereas the 28-day strength increased by 24.3%. After arriving at the goaf and completing the filling operation, the transport of excess water in the filling material slurry to the surface of the filled object removed tiny particles of cement and FA from the filled object, thereby increasing its material porosity and weakening its cementing capacity. When the concentration was larger, less water was transported, the loss of tiny particles was reduced, and the compactness of the filled object was greater, thereby giving greater strength at the macroscopic level.

*3.2. Feasibility Analysis of Cemented Filling Material Based on Aeolian Sand*

From practical engineering projects and the geological conditions of the 115 m deep filling mining face of the no. 3 coal seam in the Jinniu coal mine of Yulin in Shaanxi Province, China, the following indices of the transportation properties of cemented filling mining in this mine were obtained: a bleeding rate of less than 5% and a slump of no less than 100 mm. The following index of the mechanical properties was also obtained, namely a 28-day strength of more than 2.88 MPa. These indices ensure that the filling slurry is transported to the goaf successfully and the formed filling object has a good filling effect [22].

A comparative analysis of the results of the proportioning tests on the AS-based cemented filling material showed that the 28-day strength (minimum: 3.82 MPa) and slump (minimum: 177 mm) were good for all proportions, whereas only the D3 experimental proportions (Table 2) satisfied the required bleeding rate. Therefore, the optimal experimental proportions are at a concentration of 76%, a cement

content of 12.5%, an FA content of 47.5%, an LS content of 5%, and an AS content of 35%. The specimen with these proportions had a bleeding rate of 4.83%, a slump of 177 mm, and a 28-day strength of 11.13 MPa, which satisfies the transportation and mechanical properties required for filling mining in this particular mining area.

### 3.3. Developing Characteristics of Early Microstructure in Filled Object

The hydration of the filling material is a progressive process. As the hydration reaction proceeds, the raw filling material is consumed continually and generates products of various types and forms that change the internal structure of the filled object, influencing its mechanical properties. During the present tests, SEM was used to observe the microstructure and surface morphology of the filled objects, prepared with the optimal proportions (D3) for curing times of 3 and 7 days. Additionally, quantitative and qualitative analyses were performed on the elementary composition and distribution inside these filled objects using an EDS.

3.3.1. Results of SEM Analysis

Filled Object for a Curing Time of 3 Days

Figure 13 shows micrographs of the microstructure inside the filled object for a curing time of 3 days under various magnifications. Under 100× magnification, some voids with diameters ranging from 100 to 300 μm were observed inside the filled object, in which sand particles and cementing material were bonded together. Under 500× magnification, several cracks ranging from 50 to 200 μm in length were observed between the sand particles and cementing material. Under 2000× magnification, a few needle-like hydration products were observed on the surfaces of the sand particles. Under 6000× magnification, abundant interlaced needle-like hydration products and some flocculent hydration products were observed. The latter hydration products were distributed loosely at large intervals of 3–10 μm.

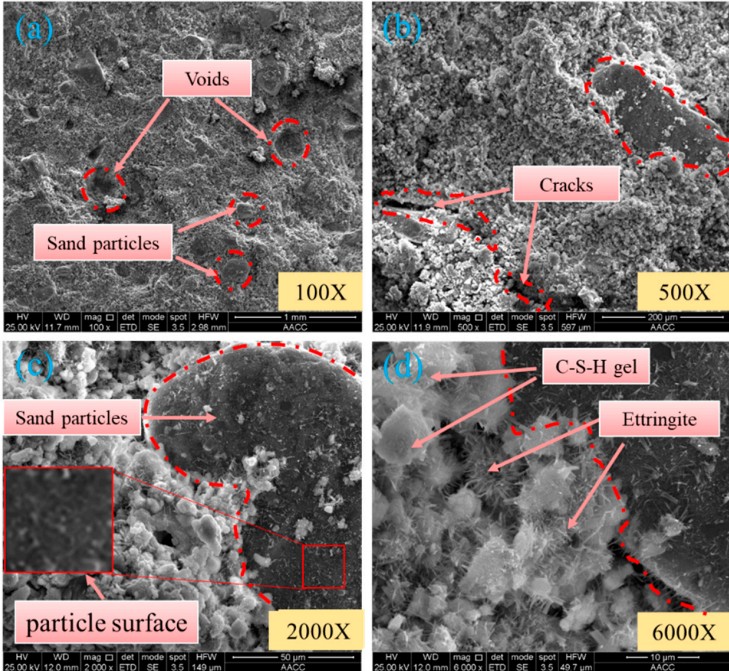

**Figure 13.** SEM micrographs of microstructure inside filled object for a curing time of 3 days under various magnifications (as indicated at the bottom right-hand corners): (**a**) filled object at 100× magnification; (**b**) filled object at 500× magnification; (**c**) filled object at 2000× magnification; (**d**) filled object at 6000× magnification.

Filled Object for a Curing Time of 7 Days

Figure 14 shows micrographs of the microstructure inside the filled object for a curing time of 7 days under various magnifications. Under 100× magnification, a small number of voids were observed inside the filled object and the integral compactness of the filled object was good. Under 500× magnification, the sand particles and cementing material bonded well, and no large cracks were observed. Under 2000× magnification, many needle-like hydration products were observed on the surfaces of the sand particles, as shown in Figure 14c. There were FA particles with a diameter of about 5 μm inside the cementing material, with an abundance of needle-like hydration products but no fracture zones on the particle surfaces. This indicated that the FA was not hydrated and could absorb the hydration products of the cement, as shown in Figure 14c. Under 6000× magnification, there was an abundance of developed flocculent hydration products stacked on top of each other inside the filled object, the compactness of the filled object was good, and there were fewer needle-like hydration products.

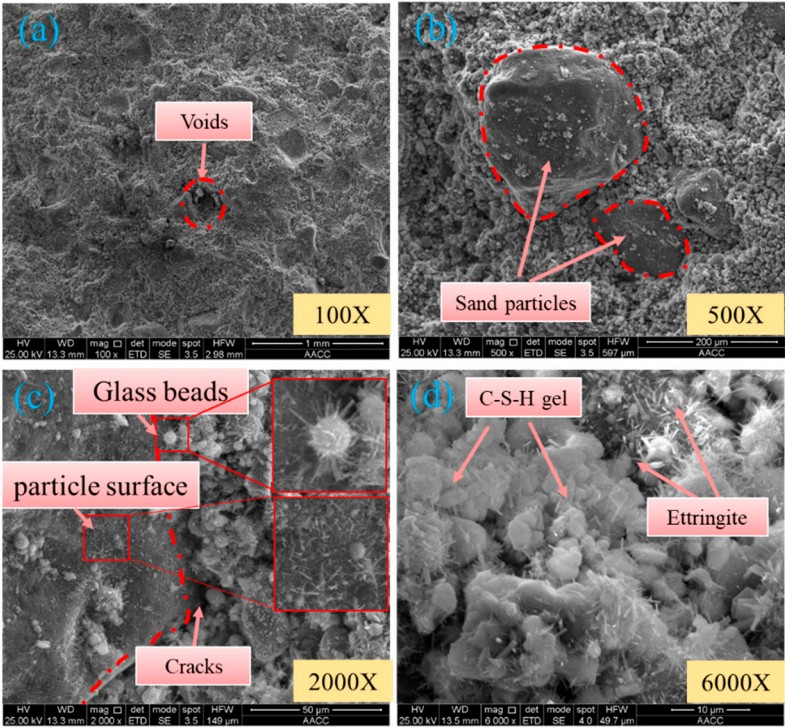

**Figure 14.** SEM micrographs of microstructure inside filled object for a curing time of 7 days under various magnifications (as indicated in the bottom right-hand corners): (**a**) filled object at 100× magnification; (**b**) filled object at 500× magnification; (**c**) filled object at 2000× magnification; (**d**) filled object at 6000× magnification.

The results obtained by observing the filled objects for curing times of 3 and 7 days using SEM show that the hydration products of the filled objects are primarily comprised of randomly distributed amorphous flocculent tobermorite as well as interlaced fiber-like and needle/rod-like ettringite [19,34]. Upon increasing the curing time from 3 to 7 days, the hydration reaction persisted as indicated by the gradual decrease in the abundance of needle-like hydration products and the gradual increase in the abundance of flocculent hydration products. The flocculent hydration products were situated on top of each other and filled the tiny voids, thereby increasing the compactness of the filled objects.

3.3.2. Results of Energy Dispersive Spectrometer (EDS) Analysis

To determine the various types of hydration products inside the filled objects for different curing times of 3 and 7 days and how their contents varied, EDS spot element testing was performed on the

filled objects for a curing time of 7 days under $1000\times$ magnification and EDS surface element testing was performed on the filled objects for curing times of 3 and 7 days.

### 3.3.3. EDS Spot Element Testing of Hydration Products in Filled Objects for a Curing Time of 7 Days

Figure 15a,b show the locations of the selected EDS testing spots for the hydration products for a curing time of 7 days, while Figure 15c,d show the testing results for elements at the selected testing spots. Given that O, Ca, Si, and Al were the four primary elements in the hydration products, the above elements were selected as the analysis objects. The testing results show that O and Ca had the highest contents in the hydration products. The contents of O and Ca in the flocculent hydration products exceeded those in the needle-like hydration products, whereas the contents of Si and Al were lower. The flocculent hydration products contained only 0.65 wt. % Al, whereas the needle-like hydration products contained 6.3 wt. % Al. According to the mechanism underlying the hydration reaction of cement [35,36], it could also be determined that the flocculent hydration product containing only a small amount of Al was C-S-H, and the needle-like hydration product was AFt.

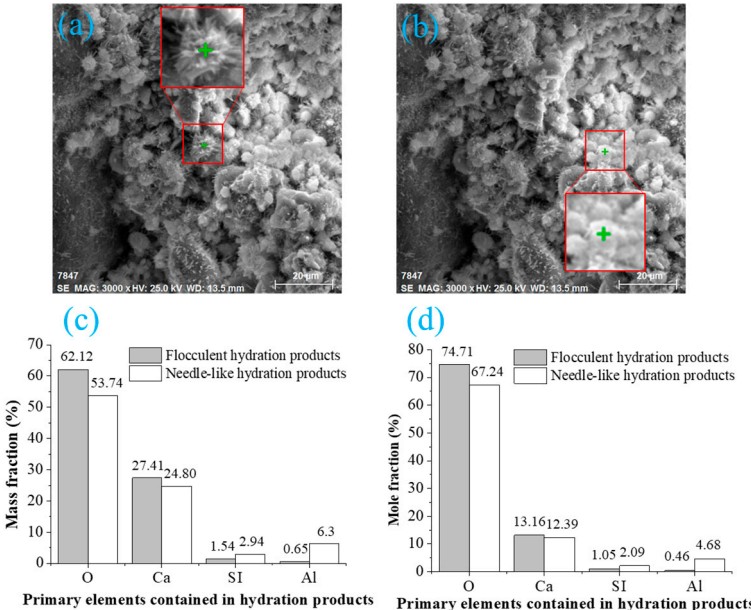

**Figure 15.** Locations and testing results of selected spots for energy-dispersive spectroscopy spot element testing of hydration products for a curing time of 7 days: (**a**) locations of testing spots in needle-like hydration products; (**b**) locations of testing spots in flocculent hydration products; (**c**) concentrations (wt. %) of elements contained in hydration products; and (**d**) mole fractions (at. %) of elements contained in hydration products.

### 3.3.4. EDS Surface Element Testing of Hydration Products in Filled Objects for Various Curing Times

Figure 16a,d show the locations and testing results for selected surfaces during EDS surface element testing of hydration products for curing times of 3 and 7 days. According to the element distributions in the testing surfaces in the filled objects with the two curing times, as shown in Figure 16b,e, the element distributions were related to the compositions inside the filled objects. Elements O, Ca, Si, and Al were distributed intensively in the region in which the hydration products were concentrated. The Si content was larger in areas where there were sand particles and there was relatively more Al, O, and Si inside the glass beads, due to the presence of $Al_2O_3$ and $SiO_2$. Analyzing Figure 16c,f allowed us to determine that the Ca content inside the filled objects increased moderately with an increase in the curing time, while the Al content decreased moderately. This indicates that during the hydration reaction, the C-S-H content increased gradually, the AFt content decreased

gradually, and the generation and transformation of hydration products in local regions influenced the transmission and redistribution of elements inside the filled objects.

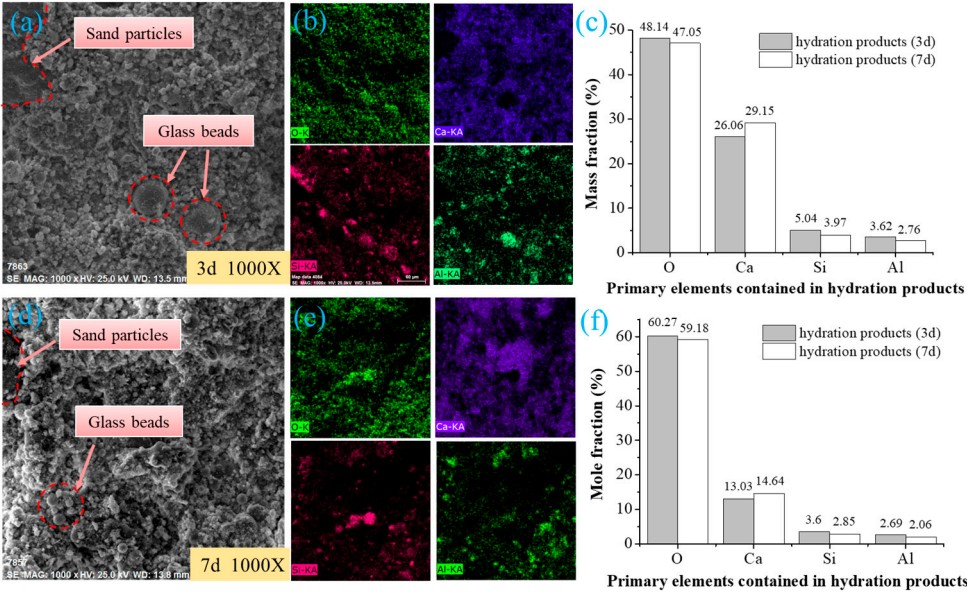

**Figure 16.** Locations and testing results for selected surfaces for EDS surface element testing of hydration products with various curing times: (**a**) testing surfaces of energy spectrum for a curing time of 3 days; (**b**) element distributions in testing surfaces for a curing time of 3 days; (**c**) element contents in testing surfaces (wt. %); (**d**) testing surfaces of energy spectrum for a curing time of 7 days; (**e**) element distributions in testing surfaces for a curing time of 7 days; and (**f**) element contents in testing surfaces (at. %).

## 4. Conclusions

(1) The feasibility of the AS-based cemented filling materials was discussed through laboratory proportioning tests. The optimal laboratory proportions for the filling material that satisfied the cemented filling mining of coal mines were determined to be a concentration of 76%, an FA content of 47.5%, a cement content of 12.5%, an LS content of 5%, and an AS content of 35%. The values of the specific indices were a bleeding rate of 4.83%, a slump of 177 mm, and a 28-day strength of 11.13 MPa.

(2) The aeolian sand-based cemented filling material has a superior transportation performance (minimum slump: 177 mm); however, the bleeding rate is poor (minimum bleeding rate of 4.83%). Upon increasing the concentration, the FA content, cement content, LS content, bleeding rate, and slump of the filling material slurry decreased gradually.

(3) The aeolian sand-based cemented filling material has excellent mechanical properties (minimum 28-day strength: 3.82 MPa), which is related to FA content, cement content, LS content, and curing time. For a constant curing time, the strength of the filled object increased gradually with an increase in the concentration, FA content, cement content, and LS content. For constant proportions, the strength of the filled object increased gradually with a longer curing time.

(4) The early microstructures of filled objects prepared using the optimal proportions were analyzed through SEM and EDS. The results show that flocculent C-S-H and needle/rod-like AFt were the primary early hydration products in the filled objects. Upon increasing the curing time from 3 to 7 days, the AFt content decreased gradually, the C-S-H content increased gradually, and the compactness of the filled objects also increased gradually.

**Author Contributions:** Conceptualization, N.Z.; Methodology, H.M. and D.G.; Formal Analysis, N.Z. and S.O.; Investigation, T.H. and D.G.; Data Curation, D.G.; Writing-Original Draft Preparation, S.O.; Writing-Review & Editing, N.Z.

**Funding:** This research was supported by the Fundamental Research Funds for the Central Universities (Grant Nos.: 2017QNA22).

**Acknowledgments:** We are grateful to the China University of Mining and Technology for providing us with the experimental platform and all the reviewers for their specific comments and suggestions.

**Conflicts of Interest:** The authors declare no conflict of interest.

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
