# Peer review of "Influential Factors in Transportation and Mechanical Properties of Aeolian Sand-Based Cemented Filling Material"

_minerals, doi:10.3390/min9020116_

Round 1
Reviewer 1 Report
Is O.K. but In table 1 show the concentration (%).
Author Response
Point 1: In table 1 should show the concentration (%)`
Response 1: Thank you for your reminder on this problem. (1) We have changed “solid mass fraction” to “concentration”; (2) The manuscript has undergone English language editing by MDPI.

Reviewer 2 Report
Overall nice paper. Needs some extra information, as indication.

Author Response
Point 1: Moderate English changes required.
Response 1: Thank you for your reminder on this problem. The manuscript has undergone English language editing by MDPI.
Point 2: Not sure how come the cumulative volume is starting at 0 or a while.
Response 2: Thank you for your reminder on this problem. The red line in Figure 2 show the cumulative volume of aeolian sand, the black line show the volume of aeolian sand; since the minimum particle size of aeolian sand is 50 um, both of them start to increase from 50 um (The minimum particle size of Fly ash and cement are 3.5um and 1.7 um, respectively).
Point 3: Please clarify why lime slag can excite fly ash activity.
Response 3: Thanks for your advice. By summarized the results of other scholars on cemented filling materials, we found that the active substances SiO2 and Al2O3 contained in fly ash can react with OH- in the alkaline solution, Si-O-Si, Al-O-Al will be destroyed, and promoting the hydration reaction of the mixed materials [1-3]. Therefore, quicklime was added to excite fly ash activity.
Point 4: Not clear how these indexes were obtained?
Response 4: Thank you for your advice.
(1) According to the GB/T50080-2016 Chinese national standard for test method of performance on ordinary fresh concrete (China Academy of Building Research), we get the slump indices of aeolian-sand-based cemented filling material should more than 100mm;
(2) By summarized the results of other scholars on cemented filling materials, we get the bleeding rate of aeolian-sand-based cemented filling material should less than 3% and 28-day strength more than 2.88Mpa [3-5].
These indices ensure that the filling slurry is transported to the goaf successfully and the formed filling object has a good filling effect.
References
[1] Wang, D.; Olowokere, D.; Zhang, L. Interpretation of Soil–Cement Properties and Application in Numerical Studies of Ground Settlement Due to Tunneling Under Existing Metro Line. Geotechnical & Geological Engineering 2014, 32, 1275-1289.
[2] Deng X, Zhang J, Klein B, Zhou N, deWit B. Experimental characterization of the influence of solid components on the rheological and mechanical properties of cemented paste backfill. Int J Miner Process. 2017; 168: 116-25.
[3] Yu Y, Development for the new backfilling cementing materials and research on its properties in coal mine[D]. China University of Mining & Technology (Beijing). 2017.
[4] Zhao C. Study on coal mine new past filling material properties and its application [D]. China University of Mining & Technology. 2017.
[5] Zhao Y. Study on the transportation properties of cement back filling slurry with high concentration Xinyang coal mine[D]. China University of Mining & Technology. 2017.

Round 2
Reviewer 2 Report
Thanks for addressing my comments. Once again, good study was conducted.
Author Response
Thank you for your reminding of English spelling. This manuscript has been polished by the English editor of MDPI.

This manuscript is a resubmission of an earlier submission. The following is a list of the peer review reports and author responses from that submission.
Round 1
Reviewer 1 Report
Dear Authors,
Please see the attached file.
Kind regards,
Reviewer

Reviewer 2 Report
The situation is rather exceptional: a mine, or better a mining area in western China, without enough gangue so that the backfilling must be performed using external materials like fly ashes, cement or lime-slags. This fact is partially explained by the fact that the exploitable mineral is coal. Nevertheless, one can be suspicious that mining extraction uses a low cut-off grade, or that the geomorphology of the ore-bodies allows a clear distinction between ore and gangue allowing for an active discrimination during extraction. The cost of transportation of alternative materials, like the ones tested, would normally became prohibitive. It is understandable that coal-based materials do not constitute a consistent material for backfilling and this type of mines probably cannot produce alternative materials.
Otherwise, the research is consistent: the mechanical and the transportation properties of each one of the alternative materials are tested, using similar and robust methodologies. The article is well written using a depurated style.
I recommend the publication of the article in its present final form.
Reviewer 3 Report
The methods are not described and only one analysis technique is used.
It must be corrected:
Line 69: calcites or calcite ?. Indicate how the minerals are identified. It is essential the mineral analysis by XRD in powder and in aggregates oriented to identify the minerals of the clay.
Methods: nothing is said about the instruments used in the determination: make, model, conditions
Figure 2: Indicate on the photo the minerals present
Lines 80 and 87: mullites or mullite?
Figures 12 and 13 c and d: SI or Si?